# Acceptability of an extended duration vaginal ring for HIV prevention and interest in a multi-purpose ring

Marie C. D. Stoner[1]*, Erica N. Browne[1], Holly M. Gundacker[2], Imogen Hawley[1], Beatrice A. Chen[3], Craig Hoesley[4], Rachel Scheckter[5], Jeanna Piper[6], Devika Singh[3], Mei Song[3], Albert Liu[7,8], Ariane van der Straten[1,9]

1 Women's Global Health Imperative, RTI International, Berkeley, CA, United States of America, 2 Statistical Center for HIV/AIDS Research & Prevention, Fred Hutchinson Cancer Research Center, Seattle, Washington, United States of America, 3 Magee-Womens Research Institute, University of Pittsburgh Medical Center, Pittsburgh, PA, United States of America, 4 University of Alabama at Birmingham, Birmingham, AL, United States of America, 5 FHI 360, Durham, NC, United States of America, 6 US National Institutes of Health, Bethesda, MD, United States of America, 7 Bridge HIV, San Francisco Department of Public Health, San Francisco, CA, United States of America, 8 Department of Medicine, University of California, San Francisco, San Francisco, CA, United States of America, 9 Center for AIDS Prevention Studies, University of California San Francisco, San Francisco, CA, United States of America

* mcstoner@rti.org

**Data Availability Statement:** Study data are available upon request from the Microbicide Trials Network by submission of a Dataset Request Form available at http://www.mtnstopshiv.org/resources.

## Abstract

Given challenges with adherence to existing HIV prevention products, the development of an extended duration vaginal ring could improve adherence while reducing patient and provider burden. Additionally, women have other interlinked sexual health concerns such as unintended pregnancy. We evaluated acceptability of a 90-day ring to prevent HIV and hypothetical preferences for a dual (HIV and contraceptive) indication. This was a secondary analysis of a Phase 1, two-arm, multi-site, placebo-controlled randomized trial evaluating safety and pharmacokinetics of a 90-day vaginal ring containing tenofovir for HIV prevention (N = 49). We used a mixed methods approach to assess quantitative data on acceptability (n = 49) and used qualitative data from a random subset to explain the quantitative findings (N = 25). The 3-month extended duration tenofovir ring was highly acceptable. Participants perceived the ring to be easy to use, comfortable and reported liking it more over time. About half felt the ring during sex but most of those participants said it bothered them only a little. Concerns about hygiene increased over the study period but were often outweighed by the benefits of an extended duration ring. Interest in a multi-purpose ring was high (77%) and even higher among those who were sexually active and had male partners. The 3-month extended duration tenofovir ring for HIV prevention was highly acceptable among women and interest in an MPT was high.

Interested parties would be able to access these data in the same manner as the authors. The authors did not have any special access privileges that others would not have.

**Funding:** The MTN-038 study was designed and implemented by the Microbicide Trials Network (MTN) funded by the National Institute of Allergy and Infectious Diseases through individual grants (UM1AI068633, UM1AI068615, UM1AI106707), with co-funding from the Eunice Kennedy Shriver National Institute of Child Health and Human Development and the National Institute of Mental Health, all components of the U.S. National Institutes of Health. The study investigators are thankful to CONRAD, which provided the rings with funding (AID-OAA-A-14-00010 and AID-OAA-A-14-00011) from the US Agency from International Development (USAID) and PEPFAR. The content is solely the responsibility of the authors and does not necessarily represent the official views of the National Institutes of Health or other agencies. Dr. Chen receives research grants from Medicines360 and Sebela, which are all managed by Magee-Womens Research Institute. The funder provided support in the form of salaries for authors BC, but did not have any additional role in the study design, data collection and analysis, decision to publish, or preparation of the manuscript. The specific roles of these authors are articulated in the 'author contributions' section.

**Competing interests:** Dr. Chen receives research grants from Medicines360 and Sebela, which are all managed by Magee-Womens Research Institute. All other authors declare no conflict of interest. This does not alter our adherence to PLOS ONE policies on sharing data and materials.

## Introduction

Women represent 48% of new human immunodeficiency (HIV) infections occurring worldwide [1]. Given the high burden in women, there is a need to develop effective and acceptable biomedical interventions to prevent HIV. Clinical trials support safety and tolerability of tenofovir (TFV) for the prevention of HIV acquisition, specifically in vaginal gel [2–4] and oral tablet formulations [5]. However, adherence has been a significant barrier to the effective use of these products, particularly among young women (age 15–24) [2, 3, 5–7]. Similarly, two phase III HIV-1 prevention trials using the dapivirine (DPV) vaginal ring reported a significant reduction in HIV-1 incidence but women under 21 years old achieved less protective benefit from the DPV ring due to low adherence [8, 9]. Given these challenges, women may benefit from increased choices in HIV prevention products to identify one that is best suited for their needs. The development of an extended duration (90 day) vaginal ring may allow less frequent ring replacements (i.e., quarterly instead of monthly) thus may improve product adherence and reduce patient and provider burden.

Additionally, women have other simultaneous and interlinked sexual health concerns such as acquisition of other sexually transmitted infections (STI) like herpes simplex virus type 2 (HSV-2), which can increase risk of HIV-1 infection [10–12], and unintended pregnancy. Multi-purpose prevention technologies (MPTs) are products that offer protection against multiple outcomes, such as multiple STIs (e.g., HSV-2 and HIV-1) or unintended pregnancy, with the use of a single product. There is a need to assess interest and preferences for an MPT among women to inform the development of these products.

The MTN-038 trial was a collaboration between the Microbicide Trials Network (MTN) and CONRAD, a not-for-profit research organization that developed the tenofovir vaginal ring, to evaluate the pharmacokinetics and safety of an extended duration (90 day) tenofovir ring, as compared to a placebo ring. We used data from MTN-038 to describe acceptability of a three-month vaginal ring (placebo versus tenofovir) for HIV prevention and hypothetical preferences for an MPT product using a mixed methods approach. While previous research has evaluated acceptability of a one-month ring, we add to the literature by describing acceptability of a three-month tenofovir ring. Specifically, our objectives were to: 1) evaluate components of acceptability of the ring used continuously for 91 days to prevent HIV in women living in three cities in the United States; and 2) understand interest/preference in a single (HIV prevention) vs. dual-purpose (contraceptive and HIV prevention) indication.

## Materials and methods

### Study population

We used quantitative and qualitative data from people who participated in the MTN-038 trial in Birmingham, AL, San Francisco, CA, and Pittsburgh, PA, between January 2019 and August 2019 to assess acceptability of the extended duration (90 day) tenofovir ring and preferences for an MPT ring. MTN-038 was a Phase 1, randomized-controlled trial to evaluate pharmacokinetics and safety of a 90-day 1.4-gram tenofovir vaginal ring to prevent HIV. The 1.4 g TFV ring consists of a drug-loaded hydrophilic polyether urethane tube (white segment) that is sealed and joined together (transparent joint) to form the shape of a ring. This is a reservoir ring using a water-absorbable polyurethane as a rate controlling membrane which can deliver approximately 10 mg/day TFV for 90 days (Fig 1). The TFV IVR has a 0.7 mm wall thickness, 5.5 mm outer cross-sectional diameter and 55 mm outer diameter. The dapivirine ring is slightly different and is a silicone polymer matrix-type ring with a cross-sectional diameter of 7.7 mm and outer diameter of 56 mm. A comparison of rings is available on the MTN 038 website [13]. The

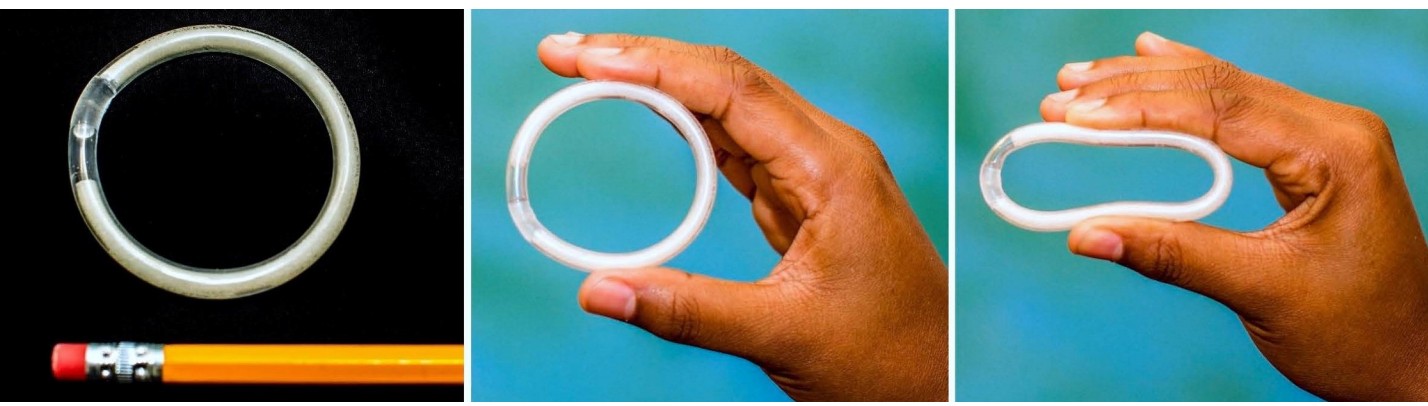

**Fig 1. CONRAD 90-day 1.4-gram tenofovir vaginal ring to prevent HIV used in the MTN 038 study.**

study enrolled 49 healthy, HIV-uninfected individuals assigned female sex at birth (inclusive of transgender men and nonbinary people), who were 18–45 years old. Sample size calculations were based upon the size of similar Phase 1 studies of vaginal microbicide products and focused on the primary endpoints of safety and pharmacokinetics of the ring. Assuming a standard deviation of 2.4. the acceptability score from a prior study, we estimated the study would have 90% power to detect a difference in the acceptability score of 2.65 with 32 participants [14]. Additional detail on sample size calculations will be available in the primary publication from the study evaluating safety and pharmacokinetics of the ring (Microbide Trials Network 038 study, 2021) [15]. All participants were required to be using an effective method of contraception (hormonal methods, intrauterine device, sterilization, sex exclusively with individuals assigned female sex at birth, or sexually abstinent) for at least 30 days prior to enrollment and intending to continue use of an effective method for the duration of study participation. Participants were randomized 2:1 to receive either the tenofovir ring or a matching placebo ring and were instructed to use the ring continuously for 91 days, including during menses. Participants in the trial were instructed not to remove the ring for 90 days after insertion including during menses, to clean it, or for intercourse. Participants were not told their group assignment to minimize bias in self-reported measures. Prior participation in other ring trials was allowed and 19 participants previously participated in MTN-036/IPM 047, a Phase 1, randomized, three-arm, open label trial that compared a 1-month vaginal ring containing 25mg to a 3-month vaginal ring containing 100mg or 200mg of dapivirine (DPV) and was conducted in Birmingham, AL and San Francisco, CA between November 2017 and January 2019.

## Procedures and measures

**Quantitative.** We collected quantitative behavioral data via computer-assisted self-interview (CASI) at enrollment, and then approximately monthly at day 28, day 56, and day 91 or product use end visit (PUEV). We assessed acceptability at day 28 and PUEV, with a more thorough questionnaire administered at PUEV. Acceptability questions were derived from an acceptability framework [16] and included participant's attitude related to the ring (ring characteristics; likes and dislikes concerning the ring), her experiences using the ring (e.g., genitourinary discomfort, ease of use/removal, willingness to use during menstruation, willingness to use in the future), and effect on sex. We assessed HIV prevention versus MPT preference only at PUEV. Other topics in the behavioral assessment were participant demographics, motivation to join the trial, vaginal and sexual practices, study product adherence, and sexual partners. All measures were used previously in MTN-036 and other microbicide trials [14, 17, 18].

We used the measures listed in Table 1 to assess components of acceptability for the extended duration ring, and ring preferences. Response options were on a 4–5 point Likert scale unless otherwise noted. All questions were asked in relation to the time period since the start of the study. For the second research objective, we assessed preference for HIV prevention vs. an MPT ring using the question "Would you be more likely to use a ring for HIV prevention if it could also prevent pregnancy (a dual-purpose ring)?" (Yes/No-equally likely/No-less likely to use MPT). Characteristics evaluated in relation to MPT preference included

**Table 1. Selected quantitative measures of acceptability and preference.**

| Domain | Measure* | Response options | Timepoint |
|---|---|---|---|
| Acceptability | Overall, how much do you like the ring? | 10-point Likert scale | Day 28 & Day 91/ PUEV |
| | | 1 = Extremely Dislike, 5 = Neutral, 10 = Extremely Like | |
| | How do you like the ring now compared to when you started the study? | I like it MORE now, | Day 28 & Day 91/ PUEV |
| | | I like it LESS now, | |
| | | I like it the SAME, | |
| | | I do not like the ring | |
| | How did it feel to have the ring inside you every day? | Very comfortable, comfortable, | Day 28 & Day 91/ PUEV |
| | | uncomfortable, very uncomfortable | |
| | How easy or difficult was it to use the ring? | Very difficult, difficult, easy, very easy | Day 28 & Day 91/ PUEV |
| | Were you aware of the ring during your normal daily activities? | Never, some of the time, most of the time, all of the time | Day 28 & Day 91/ PUEV |
| | Have you noticed any of the following changes in your vagina while wearing the ring?<br>• Vagina was wetter<br>• Vagina was drier<br>• Your vagina had a change in odor or scent | Yes, no | Day 28 & Day 91/ PUEV |
| | | Not at all, a little, somewhat, very much | |
| | How much has the change bothered you? | | |
| | How worried are you about the ring being dirty or unhygienic? | Not at all, a little, somewhat, very much | Day 28 & Day 91/ PUEV |
| | How worried are you about the ring causing infection, infertility, or other long term health problems? | Not at all, a little, somewhat, very much | Day 28 & Day 91/ PUEV |
| | How much did it bother you to wear the ring during menses? | 10-point Likert scale | Day 28 & Day 91/ PUEV |
| | | 1 = Not at all, 10 = Very much | |
| | How often did you feel the ring inside you when you had sex? | Never, some of the time, most of the time, all of the time, | Day 28 & Day 91/ PUEV |
| | How much did it bother you? | I never had sex with the ring in | |
| | | Not at all, a little, somewhat, very much | |
| Preferences | Which would you prefer: a ring that can be worn for three months and then replaced with a new one, or one that must be replaced with a new one every month? | Prefer 3 months, | Day 91/PUEV |
| | | Prefer 1 month, | |
| | | No preference | |
| | Which would you prefer: a ring that you leave in continuously, or a ring that you insert only on the days when you have sex? | Prefer leaving in continuously, | Day 91/PUEV |
| | | Prefer inserting only when I have sex, | |
| | | No preference | |
| | Which do you think your primary partner would prefer as a method to prevent HIV? | Ring, | Day 91/PUEV |
| | | Condom, | |
| | | PrEP (daily oral pill), | |
| | | Don't know, | |
| | | N/A—no partner | |

* all questions were asked in relation to the time period since the start of the study

sociodemographic variables (e.g. age, education, contraceptive use), geographical site, sexual partners and activity.

**Qualitative.** A subset of 25 participants were randomly selected to complete an in-depth interview (IDI) before exiting from the trial. IDIs were conducted by one of three trained, female, qualitative interviewers using a semi-structured questionnaire guide. Interviews were conducted over the computer via video call and lasted approximately 60 minutes. Topics assessed were: challenges using study products, specifically in relation to hygiene, menses and sex; perceived benefits and barriers to ring use; and perceived method(s) preferences for HIV prevention and MPTs. Interviewers engaged in peer debriefing sessions regularly throughout the data collection period to critically reflect on interviewing techniques. All IDIs were audio recorded and transcribed by an external transcription agency, with transcripts reviewed for quality by interviewers and qualitative analysts before coding and analysis.

## Analysis

**Quantitative.** First, we compared differences between the placebo and active ring in overall acceptability to determine if study arms could be combined. Overall acceptability was measured on a ten-point Likert scale and differences were tested using a Wilcoxon rank-sum test. We identified no significant differences by arm, and therefore all acceptability analyses used aggregate data combining the study arms. Second, we used descriptive statistics to summarize sociodemographic characteristics and sexual behavior of the study participants at baseline. For categorical variables, we reported the number and percentage in each category; for continuous variables, the mean, median, standard deviation, quartiles and range (minimum, maximum). Next, we described measures of acceptability at enrollment and at each study visit. Preferences were described only at the last visit. Differences by characteristics were assessed using Chi-squared or Fisher's exact tests. Changes in outcomes were assessed over time using Poisson regression to estimate a relative risk with robust standard errors (for binary) [19] or linear regression (for continuous). We accounted for repeated measures within participants using generalized estimating equations (GEE) [20]. Lastly, we described preferences for an MPT ring overall and by the participant characteristics using descriptive statistics and tested for differences using Chi-squared or Fisher's exact tests. We used an alpha of 0.05 to determine statistical significance. Characteristics included age, study site, whether the participant had menses while using the ring, and whether (or how much) she engaged in penile-vaginal intercourse while using the ring, as well as prior history with vaginal products and contraceptive use. Prior history of vaginal products was defined as use of the NuvaRing, Estring, Femring or prior participant in trial of vaginal ring. Use of Estring and Femring were unlikely in this premenopausal population but were included to capture even the small chance of prior use. All analyses were done with Stata version 16 (16.1, StataCorp LLC, College Station, TX).

**Qualitative.** Qualitative data were analyzed to further explain key quantitative findings. We used an existing codebook that was previously developed and applied to qualitative data in a similar trial [17]. Transcripts were coded using Dedoose software v7.0.23 by a team of two analysts who met regularly during the coding process to discuss findings and intercoder discrepancies. For this analysis, we examined code reports related to acceptability and MPT interest. This included the following codes: MPT, ATTITUDES, PROS/CONS, SEX, SIDE EFFECTS, and HEALTH. Code report excerpts were organized by participant to summarize data. Summary memos were then written, identifying themes and interpreting findings.

### Ethical statement

The MTN-038 study protocol was approved by the Institutional Review Board at each study site and was overseen by the regulatory branch of the Division of AIDS (DAIDS) and MTN. All participants provided written informed consent prior to study participation and IDI participants provided further verbal consent before being interviewed.

## Results

Characteristics of the 49 participants included in this analysis are described in Table 2. The median age was 29 years with 29% under 25. Nearly 60% (n = 28) had only male sex partners in the past year; 14% (n = 7) had both male and female sex partners, and 10% (n = 5) had only female partners. Half (n = 24) had prior experience using a vaginal ring, mostly from prior participation in another vaginal ring trial (MTN-036, n = 19). Two participants (4%) did not complete all follow-up.

### General acceptability

At study exit (n = 48), the median acceptability rating was 8 out of 10 (interquartile range, [IQR] 7–9). All other acceptability outcomes are presented in Table 3. Nearly all participants

**Table 2. Characteristics of participants enrolled in MTN-038, a Phase 1, randomized-controlled trial to evaluate a 90-day 1.4-gram tenofovir vaginal ring (N = 49)[4].**

|  | N | % |
|---|---|---|
| Age - *median (range)* | 29 | (18–43) |
| 18–24 | 14 | (29) |
| Race[1] |  |  |
| White | 31 | (63) |
| Black or African American | 15 | (31) |
| Asian | 6 | (12) |
| Graduated from college | 32 | (65) |
| Hispanic or Latinx | 3 | (6) |
| *Sexual history & orientation* |  |  |
| Biological sex of vaginal sex partner(s) in past year |  |  |
| Exclusively male | 28 | (57) |
| Exclusively female | 5 | (10) |
| Both male and female | 7 | (14) |
| No vaginal sex in past year[2] | 9 | (18) |
| *Vaginal practices & products ever used* |  |  |
| Prior use of a vaginal ring[3] | 24 | (49) |
| Current contraceptive method |  |  |
| IUD | 16 | (33) |
| Oral pill | 12 | (24) |
| Injectable | 4 | (8) |
| Implant | 3 | (6) |
| None | 14 | (29) |

[1] multiple responses allowed

[2] identify as: heterosexual (N = 6), bisexual (N = 3)

[3] such as NuvaRing, Estring, Femring (N = 12) or prior participant in trial of vaginal ring (N = 19)

[4] one participant missed the day-28 visit and another missed PUEV

**Table 3. Measures of acceptability collected via CASI among participants enrolled in MTN-038 (N = 48).**

| Acceptability measures | Day 28 | | Day 91/PUEV | |
|---|---|---|---|---|
| | N | % | N | % |
| Total | 48 | (100) | 48 | (100) |
| *Overall, how much do you like the ring? (score 1–10: Extremely like)* | | | | |
| Median (IQR) | 8.5 | (5–10) | 8 | (7–9) |
| *How do you like the ring now compared to when you started the study?* | | | | |
| I like it MORE now | 11 | (23) | 18 | (38) |
| I like it LESS now | 3 | (6) | 1 | (2) |
| I like it the SAME | 34 | (71) | 28 | (58) |
| I do not like the ring | 0 | (0) | 1 | (2) |
| *How did it feel to have the ring inside you every day?* | | | | |
| Very comfortable/comfortable | 44 | (92) | 46 | (96) |
| Uncomfortable/very uncomfortable | 4 | (8) | 2 | (4) |
| *How easy or difficult was it to use the ring?* | | | | |
| Very easy/easy | 47 | (98) | 47 | (98) |
| Difficult/very difficult | 1 | (2) | 1 | (2) |
| *Were you aware of the ring during your normal daily activities?* | | | | |
| Never | 39 | (81) | 36 | (75) |
| Most/some of the time | 9 | (19) | 12 | (25) |
| *Have you noticed the following changes in your vagina while wearing the ring?* | | | | |
| Vagina was wetter | 24 | (50) | 23 | (48) |
| *How much did that bother you?*—Not at all | 13 | (27) | 12 | (25) |
| A little/somewhat | 11 | (23) | 11 | (23) |
| Vagina was drier | 4 | (8) | 2 | (4) |
| *How much did that bother you?*—Not at all | 0 | (0) | 0 | (0) |
| A little/somewhat | 4 | (8) | 2 | (4) |
| Vagina had a change in odor or scent | 7 | (15) | 9 | (19) |
| *How much did that bother you?*—Not at all | 1 | (2) | 0 | (0) |
| A little/somewhat | 5 | (10) | 9 | (19) |
| Very much | 1 | (2) | 0 | (0) |
| *How worried are you about the ring being dirty or unhygienic?* | | | | |
| Not at all | 36 | (75) | 27 | (56) |
| A little/somewhat | 11 | (23) | 21 | (44) |
| Very much | 1 | (2) | 0 | (0) |
| *How worried are you about the ring causing infection, infertility, or other long-term health problems?* | | | | |
| Not at all | 27 | (56) | 28 | (58) |
| A little/somewhat | 20 | (42) | 19 | (40) |
| Very much | 1 | (2) | 1 | (2) |
| *How much did it bother you to wear the ring during menses?* | | | | |
| Median (IQR) | 1 | (1–2) | 1 | (1–3) |
| 1 = Not at all | 20 | (42) | 18 | (38) |
| 2 to 5 | 5 | (10) | 10 | (21) |
| 6 to 10 = Very much | 4 | (8) | 2 | (4) |
| NA, did not have menses while wearing ring in past 4 weeks | 19 | (40) | 18 | (38) |
| *How often did you feel the ring inside you when you had sex?* | | | | |
| Don't know | 20 | (42) | 14 | (29) |
| Never | 19 | (40) | 18 | (38) |
| Some/most/all of the time | 9 | (19) | 16 | (33) |

(*Continued*)

**Table 3.** (Continued)

| | Day 28 | | Day 91/PUEV | |
|---|---|---|---|---|
| *How much did it bother you?*—Not at all | 5 | (10) | 6 | (13) |
| A little/somewhat | 2 | (4) | 9 | (19) |
| Very much | 2 | (4) | 1 | (2) |

found the ring easy to use (98%, n = 47) and comfortable (96%; n = 46), and most were never aware of the ring during normal daily activities (75%, n = 36). After 28 days of use, 23% (n = 11) of participants indicated they liked the ring more than at enrollment, and the average overall acceptability improved by 1.2 points (95% CI: 0.7, 1.8; p<0.001).

## Worries about side effects and hygiene

At enrollment, roughly half of participants were either a little (47%, n = 23) or somewhat (4%, n = 2) worried about the ring causing infection, infertility, or other long-term health problems. This decreased slightly to 44% at Day 28 (n = 21) and 42% (n = 20) at PUEV (relative risk [RR] of worry at PUEV compared to enrollment 0.8, 95% CI 0.6, 1.1; p = 0.19). Conversely, concerns about ring hygiene increased over the study period. At enrollment, one-quarter of participants (27%, n = 13) were worried about the ring being unhygienic. By PUEV, 44% had concerns regarding hygiene (RR 1.6, 95% CI: 1.0, 2.6; p = 0.04). Among the 21 women who were a little or somewhat worried about hygiene at Day 91/PUEV, 4 (19%) preferred to use a 1-month ring, 15 (71%) preferred a 3-month ring, and 2 (10%) had no preference. Hence, the concern about hygiene appears to not have been strong enough to deter preference away from using a 90-day ring, given most still preferred a 3-month ring. Roughly half (56%, n = 27) noticed a change in their vaginal environment. The most common change was increased wetness (48%, n = 23), although most were not bothered by the change (52%, n = 12). Eleven women reported being bothered by increased wetness and nine reported being bothered by a change in odor. The nine women who reported being a little or somewhat bothered by the vagina having a change in odor or scent were the same women who reported being a little/ somewhat bothered by the vagina being wetter. A few participants noted increased discharge (n = 3).

In qualitative interviews, participants brought up concerns about the cleanliness of the ring, perceiving that the ring would gather bacteria or cause infections over time. However, worries were minor, and most participants did not actually experience these side effects. One participant (San Francisco, 19 years old) described persistent worries by stating:

> "*let's say I have something like a, like discharge or bleeding, yeah, I kind of might attribute it to the ring, but in reality it might not be from the ring. So, yeah, it's just something that, it's like, affects my mentality if something's going on with my health issues.*"

Some participants attributed perceived risk of side effects to the ring being left inside them for an extended duration, while others raised the opposite concern where they felt taking the ring out and touching it too frequently might increase risk. Overall, these concerns were described as minor, tempered by the idea that their bodies or vaginas would "*let them know*" if something was really wrong by causing a side effect such as a yeast infection. The concerns of hygiene usually did not outweigh the benefits. As one participant (San Francisco, age 19) stated, having some concerns about cleanliness was "*worth it*" to have the convenience of an extended duration ring.

### Wearing the ring during menses and sex

Most participants were not bothered by wearing the ring during menses (60%, n = 18/30 experiencing menses); 7% (n = 2) at PUEV were moderately to very bothered (rating 6–10). By PUEV, 33% (N = 16/49) of women said they felt the ring some, most, or all of the time during sex. Among those who felt it, 56% (n = 9/16) said that it bothered them a little/somewhat and 6% (1/16) said it bothered them very much.

In qualitative interviews, participants described worries that sex would be "*disrupted*" in some way by the ring: because it would slip out, be felt by partners, or influence partners' satisfaction. Several who felt the ring during sex described it as something that they were "*aware of*" or continuously "*noticed.*" However, "*noticeability*" of the ring was not necessarily uncomfortable or painful. This is exemplified by one participant (Birmingham, age 43 years) who described "*it wasn't really a pain, I could almost just like feel like he was hitting it, it was just a little discomfort.*" Of note, two participants thought the ring increased sexual pleasure but were not exactly sure why.

On the other hand, a few noted that the ring was painful or uncomfortable for either them or their sex partners. "*Bumping*" or "*hitting*" the ring during sex was frustrating and it embarrassed some participants when the ring was felt by their partners or fell out. Importantly, despite any burdens or frustrations from wearing the ring during sex, participants noted that the benefits of using the ring "*outweigh*" these drawbacks.

### Ring preferences and interest in MPT

At PUEV, most participants indicated preference for a 90-day ring (79%, n = 38; compared to 1 month 10% or no preference 10%) and one that could be worn continuously (83%, n = 40; compared to no preference 13% or before sex 4%) as shown in Fig 2. Three-fourths of participants (77%; n = 37) indicated they would be more likely to use a vaginal ring if it also prevented pregnancy, 11 (23%) participants indicated they were equally likely to use an MPT ring,

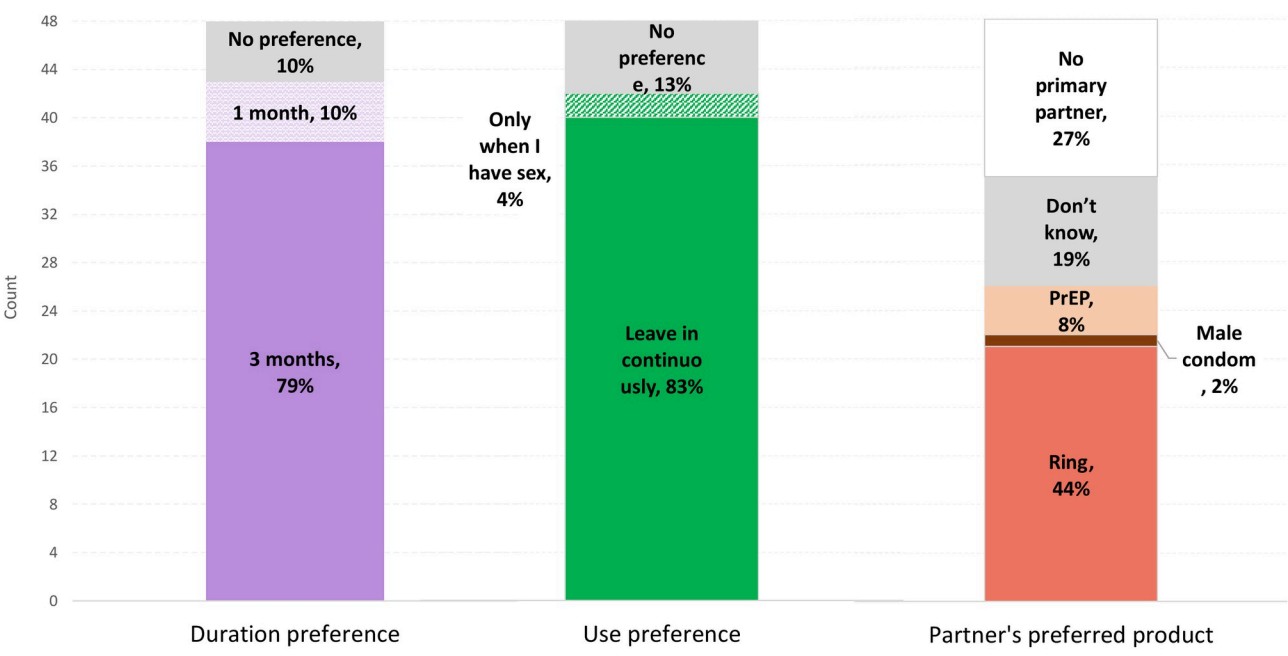

**Fig 2. Measured preferences surrounding a vaginal ring for HIV prevention at product use end visit (PUEV), N = 48.**

and no participants said they would be less likely to use an MPT ring. Many participants had a partner that preferred the ring over condoms (44%), 27% had no primary partner, 19% didn't know their partner's preferences, 8% had partners who preferred oral PrEP, and 2% had partners that preferred condoms. Trends suggested that those who had penile-vaginal sex (91%, 20/22 versus 65%, 17/26; p = 0.05), and who were using a long-acting contraceptive method at baseline (91%, 20/22 versus 65%, 17/26; p = 0.05) were more likely to be interested in an MPT ring. Age, study site, whether the participant had menses while using the ring and prior history with vaginal products were not associated with MPT interest. However, provided there were only 11 participants that were not more likely to use an MPT, it was challenging to draw conclusions about factors associated with increased likelihood of using an MPT.

In qualitative interviews, all 25 participants were supportive of an MPT ring and most thought it would be a more appealing product, described as a "*win-win*," "*super convenient*," and a "*bonus*." Some noted that they do not feel at risk of HIV personally but would consider using an MPT ring as their birth control method with the "*added benefit*" of protecting themselves from HIV "*just in case*." Another common sentiment was that an MPT may not be right for them, given their low risk of HIV and unintended pregnancy, but it would be great for others who might take birth control but forget to protect themselves from other STIs like HIV. Interestingly, that was also seen as a drawback of the MPT whereby users might think they were fully protected and fail to protect against other STIs. As stated by one participant (Pittsburgh, age 38) "*my only concern on it is that people would think that they were in the clear and it's like, no, there's a lot more STDs than just HIV*."

Several other minor concerns were noted, such as the safety of multiple drugs interacting, efficacy when combining multiple drugs and the ability to choose a hormonal contraceptive that best suits them. Ultimately, nearly all participants were supportive of an MPT ring for HIV and contraception, with the caveat that it should be safe and efficacious, and that users should also be reminded to protect themselves from other STIs.

## Discussion

Acceptability of an extended duration vaginal ring for HIV prevention in this sample of healthy, HIV uninfected US participants was generally high with participants perceiving the ring to be easy to use and comfortable. Some participants noted that they could feel the ring during sex and had concerns about hygiene that increased over time, although these concerns were generally outweighed by the benefits of the extended duration. Interest for an MPT ring was high. The only concerns that emerged about an MPT ring were that it should be safe, efficacious, and that users should be reminded to protect themselves from other STIs that are not prevented by the product. The tenofovir ring may be able to protect against HSV-2 [21], one of the most common viral STIs, which could perhaps increase further the appeal of the tenofovir ring.

General measures of acceptability were high for the extended duration tenofovir ring. This is similar to findings from trials of other formulations of the ring to prevent HIV including the one-month dapivirine vaginal ring [22], the 3 month dapivirine ring [23], and the vaginal ring containing dapivirine, maraviroc, or both [24]. In the MTN-036 trial comparing the 1-month versus the 3-month DPV ring, which nearly 40% of our sample also participated in, most users preferred the 3-month ring at the end of the study period [17]. Concerns about the extended duration ring were similar to those found in this study: worries about hygiene, potential risk of vaginal infections, and other side effects (e.g. excessive vaginal discharge, odor). In both studies, the benefits of the 3-month ring seemed to outweigh concerns, leading most participants to ultimately choose the 3-month ring in MTN-036. Our study adds to this research by

providing more data on women's experiences using a 3-month ring with a different formulation, appearance, and physical dimensions (Fig 1), but more research is needed to directly compare ring options and to allow for user choice from a variety of ring products.

Prior studies have shown that tenofovir gel can reduce incidence of HSV-2 [3, 4, 25] and earlier studies of the tenofovir ring suggested the ring could deliver enough drug to be protective against both HIV and HSV-2 [21]. Furthermore, the ring is also being evaluated with levonorgestrel for hormonal contraception, as an MPT product with multiple indications [21]. While the ring evaluated in this study may protect against both HIV and HSV-2, participants were asked about interest in an MPT ring for HIV and pregnancy prevention. Interest in an MPT product was high among all participants in this study with some minor caveats about safety, and prevention of other STIs.

Our study had several limitations. First, the sample size was small with only 49 participants. The small sample size limited the inferences about acceptability and MPT interest, particularly in relation to how preferences varied by participant characteristics. Given the small sample, we also combined placebo and active product arms based on similarities in overall acceptability between the arms. However, it is possible that some components of acceptability may have differed by arm. Second, data in our study are from a randomized trial of the ring, and there may be some social desirability bias in answers about ring preferences or acceptability whereby users may respond in a more favorable manner. Given that acceptability responses were self-reported based on experiences using the ring in the trial, we cannot determine how their choices may differ outside of this setting or how preferences were potentially influenced by social desirability or recall bias. Although, measures were captured using CASI to encourage honest feedback compared to one another. Additionally, the trial included women in the United States who were at low risk for HIV and may not reflect preferences of women at higher risk for HIV acquisition or in other countries. Future studies should be done with women at higher risk of HIV and in a variety of settings. All participants were required to be using an effective method of contraception prior to enrollment and a high percentage (33%) were using an IUD for contraception. The sample may therefore be less generalizable and may have been more likely to prefer an MPT product. Lastly, a large percentage of our sample also participated in MTN-036 and had some familiarity with using a ring in an HIV prevention study. Our sample could be biased towards liking the ring if those who may not have liked the ring in MTN-036 did not enroll in MTN-038. Sexual partners were also not interviewed and may have influenced the decision of the participants to use the ring, although participants were asked about their partner's support.

## Conclusions

The 3-month extended duration tenofovir ring was highly acceptable and acceptability increased with use, similar to findings from previous studies of vaginal rings for HIV prevention and other indications. Some participants did have worries about side effects, hygiene and wearing the ring during sex, but these concerns were mostly minor and were often outweighed by the benefits of a long-acting ring. Interest in an MPT product was high and more research is needed to allow for user choice from a variety of ring products.

## Supporting information

**S1 File. In depth interview guide used in the MTN 038 study.**
(PDF)

## Acknowledgments

We thank the study participants as well as the MTN 038 Study team members who implemented the trials.

## Author Contributions

**Conceptualization:** Marie C. D. Stoner, Erica N. Browne, Holly M. Gundacker, Imogen Hawley, Beatrice A. Chen, Craig Hoesley, Rachel Scheckter, Mei Song, Albert Liu, Ariane van der Straten.

**Data curation:** Holly M. Gundacker, Imogen Hawley, Devika Singh, Ariane van der Straten.

**Formal analysis:** Marie C. D. Stoner, Erica N. Browne, Holly M. Gundacker, Imogen Hawley, Ariane van der Straten.

**Funding acquisition:** Beatrice A. Chen, Craig Hoesley, Jeanna Piper, Albert Liu.

**Investigation:** Marie C. D. Stoner, Erica N. Browne, Holly M. Gundacker, Imogen Hawley, Beatrice A. Chen, Craig Hoesley, Rachel Scheckter, Devika Singh, Mei Song, Albert Liu, Ariane van der Straten.

**Methodology:** Marie C. D. Stoner, Erica N. Browne, Holly M. Gundacker, Imogen Hawley, Beatrice A. Chen, Craig Hoesley, Devika Singh, Mei Song, Albert Liu, Ariane van der Straten.

**Project administration:** Imogen Hawley, Beatrice A. Chen, Craig Hoesley, Rachel Scheckter, Jeanna Piper, Devika Singh, Mei Song, Albert Liu, Ariane van der Straten.

**Resources:** Beatrice A. Chen, Craig Hoesley, Jeanna Piper, Albert Liu, Ariane van der Straten.

**Supervision:** Beatrice A. Chen, Craig Hoesley, Rachel Scheckter, Jeanna Piper, Albert Liu, Ariane van der Straten.

**Writing – original draft:** Marie C. D. Stoner.

**Writing – review & editing:** Erica N. Browne, Holly M. Gundacker, Imogen Hawley, Beatrice A. Chen, Craig Hoesley, Rachel Scheckter, Jeanna Piper, Devika Singh, Mei Song, Albert Liu, Ariane van der Straten.

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
