## [Decision Letter · Decision Letter 0]

12 Nov 2021

PONE-D-21-20792Acceptability of an extended duration vaginal ring for HIV prevention and interest in a multi-purpose ringPLOS ONE

Dear Dr. Stoner,

Thank you for submitting your manuscript to PLOS ONE. After careful consideration, we feel that it has merit but does not fully meet PLOS ONE’s publication criteria as it currently stands. Therefore, we invite you to submit a revised version of the manuscript that addresses the points raised during the review process.

The manuscript has been evaluated by three reviewers, and their comments are available below. 

The reviewers feel that stronger descriptions of the medical device used in the study can further strengthen the manuscript. And, the reviewers have also provided additional discussion points for the authors to consider. 

Could you please revise the manuscript to carefully address the concerns raised?

We look forward to receiving your revised manuscript.

Kind regards,

Lucinda Shen, MSc

Staff Editor

PLOS ONE

Journal Requirements:

2. Please include a copy of the interview guide used in the study, in both the original language and English, as Supporting Information, or include a citation if it has been published previously.

” The MTN-038 study was designed and implemented by the Microbicide Trials Network (MTN) funded by the National Institute of Allergy and Infectious Diseases through individual grants (UM1AI068633, UM1AI068615, UM1AI106707), with co-funding from the *Eunice Kennedy Shriver* National Institute of Child Health and Human Development and the National Institute of Mental Health, all components of the U.S. National Institutes of Health. The study investigators are thankful to CONRAD, which provided the rings with funding (AID-OAA-A-14-00010 and AID-OAA-A-14-00011) from the US Agency from International Development (USAID) and PEPFAR. The content is solely the responsibility of the authors and does not necessarily represent the official views of the National Institutes of Health or other agencies. We thank the study participants as well as the MTN 038 Study team members who implemented the trials.”

“The MTN-038 study was designed and implemented by the Microbicide Trials Network (MTN) funded by the National Institute of Allergy and Infectious Diseases through individual grants (UM1AI068633, UM1AI068615, UM1AI106707), with co-funding from the Eunice Kennedy Shriver National Institute of Child Health and Human Development and the National Institute of Mental Health, all components of the U.S. National Institutes of Health. The study investigators are thankful to CONRAD, which provided the rings with funding (AID-OAA-A-14-00010 and AID-OAA-A-14-00011) from the US Agency from International Development (USAID) and PEPFAR. The content is solely the responsibility of the authors and does not necessarily represent the official views of the National Institutes of Health or other agencies. We thank the study participants as well as the MTN 038 Study team members who implemented the trials.”

4. Thank you for stating the following in the Competing Interests/Financial Disclosure section:

“Dr. Chen receives research grants from Medicines360 and Sebela, which are all managed by Magee-Womens Research Institute. All other authors declare no conflict of interest.”

We note that one or more of the authors are employed by a commercial company: Medicines360 and Sebela

5. Please include a caption for figure 1.

Reviewers' comments:

Reviewer's Responses to Questions

**Comments to the Author**

1. Is the manuscript technically sound, and do the data support the conclusions?

Reviewer #1: Yes

Reviewer #2: Yes

Reviewer #3: Yes

2. Has the statistical analysis been performed appropriately and rigorously? 

Reviewer #1: Yes

Reviewer #2: Yes

Reviewer #3: I Don't Know

3. Have the authors made all data underlying the findings in their manuscript fully available?

Reviewer #1: Yes

Reviewer #2: No

Reviewer #3: Yes

4. Is the manuscript presented in an intelligible fashion and written in standard English?

Reviewer #1: Yes

Reviewer #2: Yes

Reviewer #3: Yes

5. Review Comments to the Author

Reviewer #1: The paper describes user acceptability of a 90-day tenofovir-releasing vaginal ring for HIV prevention, as part of a Phase I trial. Acceptability data for a 30-day use regimen have previously been reported for the same ring device. Overall, the findings are consistent with previous reports for both this specific ring device and other vaginal ring devices. User interest in a hypothetical multipurpose ring – offering protection against both HIV infection and prevention of unintended pregnancy – was also evaluated.

The scope of the study is somewhat limited. As the authors, acknowledge, user acceptability data has already been reported for the one-month tenofovir ring, and this study simply extends the acceptability assessment out to three months. Were the researchers expecting the ring acceptability data to be different between the two use regimes? What additional information have they gleaned from this study that was unavailable from the previous study?

The study is further limited by having being conducted only in US women, rather than women in countries/communities where HIV prevalence is higher (e.g. Africa). I assume that such women are the primary market for a HIV prevention ring. Presumably, acceptability data could look very different comparing US women with African women? What was the rationale for conducting the study in US women? The authors need to comment on these issues.

Certainly, there is a need to assess user interest and preferences around an MPT ring for the purpose of informing future development of such products. However, the assessment described in the manuscript was the weakest aspect of this study, with only very limited data collected. It appears that only a single question on the topic was asked – “Would you be more likely to use a ring for HIV prevention if it could also prevent pregnancy (a dual-purpose ring)?”. A much more comprehensive study would have been helpful. For example, when presenting a hypothetical MPT ring to women combining HIV prevention and contraceptive activity, were women presented with hormonal or non-hormonal contraceptive options? Would the offer or one or the other impact user interest and preferences? Did women’s experience and satisfaction (or lack thereof) with current contraceptive use impact their preferences around an MPT ring? It was also noted that 24 (50%) of the 49 women enrolled in the study had experience with other vaginal ring devices, mostly as part of the MTN-036 trial. Did the previous experiences of these women impact their assessment and acceptability of the tenofovir ring in this study? Were the dapivirine and tenofovir rings similar in appearance and characteristics? Were any of the women currently using the contraceptive ring product NuvaRing? This would be important to know, since it would likely impact ring acceptability. I also assume that none of the women in this study were using Estring or Femring, as these are estrogen replacement products pimply indicated for us in older menopausal women; see footnote to Table 2. Surprisingly, none of these issues are documented or commented upon in the manuscript.

Did the quantitative behavioural study conducted by computer-assisted self-interview also capture information on the incidence and causes of involuntary ring expulsions? In Line 98, the authors refer to ring ‘displacement’. Does this refer to slipping of the ring within the vagina, without actual expulsion from the vagina? Please clarify.

The data collected around measures of acceptability are, of course, self-reported data, which are known to suffer from poor reliability. Any comments? For example, how do women’s reported preferences for a 90-day ring stack up against their concerns about long-term hygiene? If hygiene was an issue for some women, did these same women report a preference for the one-month ring?

PLOS ONE supports colour figures. It would have been helpful to colour code the bars in Figure 1 for ease of rating and interpreting the figure.

Reviewer #2: I thought this was really well written and a good example of using quantitative and qualitative data together (though I have very little qualitative experience so maybe I'm mistaken there). Quantitative methods-wise, this seems solid and I only have a few suggestions below to clarity and tighten a few loose bolts. Best of luck on future endeavors.

1. (lines 79-80) I find it really tough to know what to recommend for sample size calculation reporting in secondary analyses and sub-studies. I can understanding shifting these sorts of details to either a protocol publication or the main paper since usually trials were not powered for secondary analyses. Though, there's no word count for PLOS ONE and, at least for me, I regard this as pretty important information. How soon do you think the main MTN038 publication will be out? If you feel like it won't be out before this gets accepted, then I would include the sample size calculations here. If not, I'd still recommend including that info, but I'm fine if you still don't want to.

2. (line 134-135) I was a little confused about this statement because it suggests that you used Poisson regression for a binary outcome. Is this to obtain relative risks like is shown in https://doi.org/10.1093/aje/kwh090? If so, great, just please include that or another reference which explains the approach. Otherwise, maybe some editing or changes to the model are needed.

3. (line 136) Also a GEE reference here would be good.

4. For the quantitative analyses, please indicate the software and version used for analyses.

5. Although your confidence intervals say 95%, it's probably still worth noting in the quantitative methods second that you are using the 5% level of significance.

Reviewer #3: HIV prevention and reliable contraception are both important issues. Combination into one multi-purpose product would be an advantage for many women.

This study on the acceptability of a 3-month vaginal ring for HIV prevention and on the interest of the participants in a multi-purpose ring reveals interesting results for future implementation. However, several questions must be answered before acceptance.

Abstract: correct ‘form’ into ‘from’.

Introduction: no comment.

Materials and Methods

For the unfamiliar readers a short description of the ring used in the study would be very welcome.

Concerns about ring hygiene increased over the study period (Results). Was it allowed to take out the ring to clean it? If yes, please indicate for how long and how to clean? If not, can the authors discuss the pros and cons of this item in the Discussion section?

Was it allowed to take out the ring during intercourse? If yes, please indicate for how long? If not, can the authors discuss the pros and cons of this item in the Discussion section?

Results

Can the authors provide the number/percentage of partial and complete expulsions or slipping of the ring?

Table 3: at day 91/PUEV, the number of women who were bothered by ‘vagina was wetter’ and ‘vagina had a change in odor or scent’, were 11 and 9, respectively. Were these the same women? Can the authors provide data of vaginal cultures of these women? If not, discuss this in the Discussion section.

At day 91/PUEV, the number of women who felt the ring during sex some/most/all of the time was 16 (33%). These data in table 3 are not in correspondence with the text ‘By PUEV, 69% of those who reported wearing the ring during sex said they felt the ring at least some of the time (n=24/35); five (14%) felt it most or all of the time. Among those who felt it, 42% (n=10) said that it bothered them only a little (n=8). Can the authors elucidate this?

Figure 1: Insert bar with data on ‘Three-fourths of participants (77%; n=37) indicated they would be more likely to use a vaginal ring if it also prevented pregnancy, 11 (23%) participants indicated they were equally likely to use an MPT ring, and no participants said they would be less likely to use an MPT ring.’

The data of partner’s preferred product as shown in Figure 1 must also be described in the text.

Discussion

See above-mentioned items.

An important limitation of the study is that sexual partners have not been interviewed on their judgement of the ring as they may greatly influence the decision of the participants to use it. This must be discussed.

6. PLOS authors have the option to publish the peer review history of their article (what does this mean?). If published, this will include your full peer review and any attached files.

Reviewer #1: No

Reviewer #2: No

Reviewer #3: No

---

## [Author Response · Author response to Decision Letter 0]

5 Jan 2022

Reviewer's Responses to Questions

1. Is the manuscript technically sound, and do the data support the conclusions?

Reviewer #1: Yes

Reviewer #2: Yes

Reviewer #3: Yes

 RESPONSE: Thank you for this review and for your support of our analysis and interpretations.

2. Has the statistical analysis been performed appropriately and rigorously? 

Reviewer #1: Yes

Reviewer #2: Yes

Reviewer #3: I Don't Know

RESPONSE: We had added additional detail to the manuscript regarding the statistical analysis. Please see responses to review comments below.

3. Have the authors made all data underlying the findings in their manuscript fully available?

Reviewer #1: Yes

Reviewer #2: No

Reviewer #3: Yes

RESPONSE Data are available through the Microbicide Trials Network (MTN). We have also now included a copy of the qualitative interview guide. 

4. Is the manuscript presented in an intelligible fashion and written in standard English?

PLOS ONE does not copyedit accepted manuscripts, so the language in submitted articles must be clear, correct, and unambiguous. Any typographical or grammatical errors should be corrected at revision, so please note any specific errors here

Reviewer #1: Yes

Reviewer #2: Yes

Reviewer #3: Yes

RESPONSE: We have edited the manuscript again and addressed all comments below.

Review Comments to Author:

Reviewer #1: 

1.1 “The scope of the study is somewhat limited. As the authors, acknowledge, user acceptability data has already been reported for the one-month tenofovir ring, and this study simply extends the acceptability assessment out to three months. Were the researchers expecting the ring acceptability data to be different between the two use regimes? What additional information have they gleaned from this study that was unavailable from the previous study?”

RESPONSE: While similar products are available, we believe is still important to understand end user experiences and perspectives as new products are being developed because these findings will inform access and messaging when the product become available. In this case, the parent MTN 038 study was a clinical trial to assess the safety and pharmacokinetics of the three-month ring. However, we also assessed acceptability and preferences to inform how women use this product and concerns that might arise. We found that there were concerns about hygiene, sex and menses likely because of the longer duration of use. We also include a reference to the MTN 036 study comparing the dapivirine vaginal ring for 3 versus 1 month which found that users preferred 3-month rings despite having more hesitations or skepticism for the longer duration than a month-long ring.

To add to this point, we have edited the discussion section at line 290 to read as follows. “Our study adds to this research by providing more data on women’s experiences using a 3-month ring with a different formulation, but more research is needed to directly compare ring options and to allow for user choice from a variety of ring products.”

1.2 “The study is further limited by having being conducted only in US women, rather than women in countries/communities where HIV prevalence is higher (e.g. Africa). I assume that such women are the primary market for a HIV prevention ring. Presumably, acceptability data could look very different comparing US women with African women? What was the rationale for conducting the study in US women? The authors need to comment on these issues.”

RESPONSE: We agree that this ring should be available for women in all countries including the US and Africa. This study was a preliminary investigation to determine safety, pharmacokinetics and acceptability of the ring but we agree that future studies should be done in other populations. We have added this limitation to the discussion section at lines 311. “Additionally, the trial included women in the United States who were at low risk for HIV and may not reflect preferences of women at higher risk for HIV acquisition or in other countries. Future studies should be done with women at higher risk of HIV and in a variety of settings.”

1.3. “Certainly, there is a need to assess user interest and preferences around an MPT ring for the purpose of informing future development of such products. However, the assessment described in the manuscript was the weakest aspect of this study, with only very limited data collected. It appears that only a single question on the topic was asked – “Would you be more likely to use a ring for HIV prevention if it could also prevent pregnancy (a dual-purpose ring)?”. A much more comprehensive study would have been helpful. For example, when presenting a hypothetical MPT ring to women combining HIV prevention and contraceptive activity, were women presented with hormonal or non-hormonal contraceptive options? Would the offer or one or the other impact user interest and preferences?“

RESPONSE: We agree that the assessment of MPT interest and preferences is limited and should be followed up with additional research on more specific aspects of an MPT like hormonal and non-hormonal contraception options. This study was primarily descriptive and additional data were not available for analysis in this study. 

1.4. “Did women’s experience and satisfaction (or lack thereof) with current contraceptive use impact their preferences around an MPT ring? It was also noted that 24 (50%) of the 49 women enrolled in the study had experience with other vaginal ring devices, mostly as part of the MTN-036 trial. Did the previous experiences of these women impact their assessment and acceptability of the tenofovir ring in this study? Were the dapivirine and tenofovir rings similar in appearance and characteristics? Were any of the women currently using the contraceptive ring product NuvaRing? This would be important to know, since it would likely impact ring acceptability. I also assume that none of the women in this study were using Estring or Femring, as these are estrogen replacement products pimply indicated for us in older menopausal women; see footnote to Table 2. Surprisingly, none of these issues are documented or commented upon in the manuscript.”

RESPONSE: We did examine predictors of MPT preferences including age, study site, whether the participant had menses while using the ring, whether (or how much) she engaged in penile-vaginal intercourse while using the ring, as well as prior history with vaginal products and contraceptive use. In the results at line 248 we note that “Trends suggested that those who had penile-vaginal sex (91%, 20/22 versus 65%, 17/26; p=0.05), and who were using a long-acting contraceptive method at baseline (91%, 20/22 versus 65%, 17/26; p=0.05) were more likely to use an MPT ring.” We did not include full results in a table because of small samples sizes. However, in the results at line 251 we have now added “Age, study site, whether the participant had menses while using the ring and prior history with vaginal products were not associated with MPT interest. However, provided there were only 11 participants that were not more likely to use an MPT, it was challenging to draw conclusions about factors associated with increased likelihood of using an MPT” We did not measure whether participants were satisfied with their current contractive method. 

No participants were currently using a ring, Nuvaring or Estring or Femring. The footnote in Table 2 relates to whether participants had ever used a vaginal ring in the past. We have updated the text in Table 2 to “Prior use of a vaginal ring3” to clarify that it is not current use. We have also updated in the footnote to include counts of each type of ring experience “such as NuvaRing, Estring, Femring (N=12) or prior participant in trial of vaginal ring (N=19).” At line 177, we note that “Half (n=24) had prior experience using a vaginal ring, mostly from prior participation in another vaginal ring trial (MTN-036, n=19).”

Prior vaginal ring use was defined as use of the NuvaRing, Estring, Femring or prior participant in trial of vaginal ring. All vaginal rings approved for use in the United States were included in the protocol definition of vaginal ring use along with prior use of investigational vaginal rings during participation in previous research studies. Use of Estring and Femring were unlikely in this premenopausal population but were included to capture even the small chance of prior use. At line 157 in the methods we have now added “Prior history of vaginal products was defined as use of the NuvaRing, Estring, Femring or prior participant in trial of vaginal ring. Use of Estring and Femring were unlikely in this premenopausal population but were included to capture even the small chance of prior use.”

The tenofovir and dapivirine rings are similar but do have some differences. The tenofovir ring is a reservoir ring and has a 5.5 mm outer cross-sectional diameter and 55 mm outer diameter. A polyether urethane tube is sealed and joined together to form the shape of a ring. The dapivirine ring is difference and is a silicone polymer matrix-type ring and has a cross-sectional diameter of 7.7 mm and outer diameter of 56 mm. Injection molding of silicone elastomers forms the ring shape. We have added more details about the tenofovir ring in the methods at line 77, included a figure 1 with a photo of the ring and a reference to a ring comparison tool used in the MTN 038 study (See response to reviewer 3.1).

1.5. “Did the quantitative behavioural study conducted by computer-assisted self-interview also capture information on the incidence and causes of involuntary ring expulsions? In Line 98, the authors refer to ring ‘displacement’. Does this refer to slipping of the ring within the vagina, without actual expulsion from the vagina? Please clarify.”

RESPONSE: Information was captured about the incidence of ring expulsions/displacement and reasons. However, this information will be reported in the primary paper from the study and is not included here. We have edited line 112 to remove mention of displacement/expulsions.

1.6. “The data collected around measures of acceptability are, of course, self-reported data, which are known to suffer from poor reliability. Any comments? For example, how do women’s reported preferences for a 90-day ring stack up against their concerns about long-term hygiene? If hygiene was an issue for some women, did these same women report a preference for the one-month ring?”

RESPONSE: We agree with the reviewer that self-reported data can suffer from poor reliability because of social desirability or selective recall. However, acceptability is a subjective opinion which cannot be inferred entirely by clinical or laboratory assessments. For example, it is possible that the ring could cause an adverse event but that a participant may still find the ring acceptability because other benefits outweigh the concern. There are also some factors of acceptability which cannot be measured except by self-report, like how the ring feels in-situ and during intercourse. Even when clinical assessments cannot detect an issue, the participant’s perception and experience will ultimately influence their continual use of a product. Therefore, it is still valuable to collect participants’ perspectives and opinions. To encourage honest feedback and minimize social desirability bias, we measured acceptability via CASI (computer-assisted self-interviewing).

We have added the following to the results section at line 196. “Among the 21 women who were a little or somewhat worried about hygiene at Day 91/PUEV, 4 (19%) preferred to use a 1-month ring, 15 (71%) preferred a 3-month ring, and 2 (10%) had no preference. Hence, the concern about hygiene appears to not have been strong enough to deter preference away from using a 90-day ring, given most still preferred a 3-month ring.” This was also reflected in the qualitative data, as one participant shared that even with some concerns about hygiene, it was “worth it” to have the convenience of a 3-month ring. 

Given the design of the study, we only assessed experiences using the ring and preferences based on those experiences. We did not measure how these preferences compare to one another or choices that would be made in a real-world setting. The following sentence has been added to the limitations on line 308 “Given that acceptability responses were self-reported based on experiences using the ring in the trial, we cannot determine how their choices may differ outside of this setting or how preferences were potentially influenced by social desirability or recall bias. Although, measures were captured using CASI to encourage honest feedback.”

1.7. “PLOS ONE supports colour figures. It would have been helpful to colour code the bars in Figure 1 for ease of rating and interpreting the figure.”

RESPONSE: We have now color-coded the bars in figure 1 (now figure 2 due to the addition of a new figure).

Reviewer #2:

2.1. “(lines 79-80) I find it really tough to know what to recommend for sample size calculation reporting In secondary analyses and sub-studies. I can understanding shifting these sorts of details to either a protocol publication or the main paper since usually trials were not powered for secondary analyses. Though, there's no word count for PLOS ONE and, at least for me, I regard this as pretty important information. How soon do you think the main MTN038 publication will be out? If you feel like it won't be out before this gets accepted, then I would include the sample size calculations here. If not, I'd still recommend including that info, but I'm fine if you still don't want to.”

RESPONSE: Additional details on the sample size calculation has been added to the methods at line 85. “Sample size calculations were based upon the size of similar Phase 1 studies of vaginal microbicide products and focused on the primary endpoints of safety and pharmacokinetics of the ring. Assuming a standard deviation of 2.4 from the acceptability score in a prior study, we estimated the study would have 90% power to detect a difference in the acceptability score of 2.65 with 32 participants. Additional detail on sample size calculations will be available in the primary publication from the study evaluating safety and pharmacokinetics of the ring.”

2.2. “(line 134-135) I was a little confused about this statement because it suggests that you used Poisson regression for a binary outcome. Is this to obtain relative risks like is shown in https://doi.org/10.1093/aje/kwh090? If so, great, just please include that or another reference which explains the approach. Otherwise, maybe some editing or changes to the model are needed.”

RESPONSE: Thank you for providing this reference. We have added the suggested reference by Zou et al. in the methods at line 150 in the manuscript and clarified that this was done to obtain a relative risk. 

2.3. “(line 136) Also a GEE reference here would be good.”

RESPONSE: We have added a reference for the use of GEE to account for repeated measures at line 152 in the methods section.

2.4. “For the quantitative analyses, please indicate the software and version used for analyses.”

RESPONSE: The following sentence was added to the methods section at line 160. “All analyses were done with Stata version 16 (16.1, StataCorp LLC, College Station, TX).”

2.5. “Although your confidence intervals say 95%, it's probably still worth noting in the quantitative methods second that you are using the 5% level of significance.”

RESPONSE: The following sentence was added to the methods at line 154. “We used an alpha of 0.05 to determine statistical significance.”

Reviewer #3: 

 3.1. ‘Abstract: correct ‘form’ into ‘from’.”

RESPONSE: Thank you for catching this typo. “Form” has been edited to “from” in the abstract.

3.2. “Materials and Methods. For the unfamiliar readers a short description of the ring used in the study would be very welcome.”

RESPONSE: The following sentence has been added to the materials and methods at line 77 “The 1.4 g TFV ring consists of a drug-loaded hydrophilic polyether urethane tube (white segment) that is sealed and joined together (transparent joint) to form the shape of a ring. This is a reservoir ring using a water-absorbable polyurethane as a rate controlling membrane which can deliver approximately 10 mg/day TFV for 90 days (Fig 1). The TFV IVR has a 0.7 mm wall thickness, 5.5 mm outer cross-sectional diameter and 55 mm outer diameter. The dapivirine ring is slightly different and is a silicone polymer matrix-type ring with a cross-sectional diameter of 7.7 mm and outer diameter of 56 mm. A comparison of rings is available on the MTN 038 website.[13]”

3.3. “Concerns about ring hygiene increased over the study period (Results). Was it allowed to take out the ring to clean it? If yes, please indicate for how long and how to clean? If not, can the authors discuss the pros and cons of this item in the Discussion section? Was it allowed to take out the ring during intercourse? If yes, please indicate for how long? If not, can the authors discuss the pros and cons of this item in the Discussion section?”

RESPONSE: This is a good point. The following sentence has been added to the methods at line 97 to clarify. “Participants in the trial were instructed not to remove the ring for 90 days after insertion including during menses, to clean it, or for intercourse.”

3.5. “Results: Can the authors provide the number/percentage of partial and complete expulsions or slipping of the ring?”

RESPONSE: This information will be included in the primary manuscript from the study and is not reported in this manuscript. 

3.6. “Table 3: at day 91/PUEV, the number of women who were bothered by ‘vagina was wetter’ and ‘vagina had a change in odor or scent’, were 11 and 9, respectively. Were these the same women? Can the authors provide data of vaginal cultures of these women? If not, discuss this in the Discussion section.”

RESPONSE: Yes, the nine women who reported being a little or somewhat bothered by the vagina having a change in odor or scent also reported being a little/somewhat bothered by the vagina being wetter. The following sentences have been added at line 202. “Eleven women reported being bothered by increased wetness and nine reported being bothered by a change in odor. The nine women who reported being a little or somewhat bothered by the vagina having a change in odor or scent were the same women who reported being a little/somewhat bothered by the vagina being wetter.” We are unable to present data on vaginal cultures because these data were not available at the time of this analysis and will be available in future publications from the study. 

3.7. “At day 91/PUEV, the number of women who felt the ring during sex some/most/all of the time was 16 (33%). These data in table 3 are not in correspondence with the text ‘By PUEV, 69% of those who reported wearing the ring during sex said they felt the ring at least some of the time (n=24/35); five (14%) felt it most or all of the time. Among those who felt it, 42% (n=10) said that it bothered them only a little (n=8). Can the authors elucidate this?”

RESPONSE: Thank you for catching this issue. The text has now been edited in the results at line 224 to read “By PUEV, 33% (N=16/49) of women said they felt the ring some, most, or all of the time during sex. Among those who felt it, 56% (n=9/16) said that it bothered them a little/somewhat and 6% (1/16) said it bothered them very much. “

3.8. “Figure 1: Insert bar with data on ‘Three-fourths of participants (77%; n=37) indicated they would be more likely to use a vaginal ring if it also prevented pregnancy, 11 (23%) participants indicated they were equally likely to use an MPT ring, and no participants said they would be less likely to use an MPT ring.’ The data of partner’s preferred product as shown in Figure 1 must also be described in the text.”

RESPONSE: The following sentence has been added to the results section at line 246 “Most participants had a partner that preferred the ring (44%), 27% had no primary partner, 19% didn’t know their partner’s preferences, 8% had partners who preferred PrEP, and 2% had partners that preferred condoms.”

3.9. “Discussion. An important limitation of the study is that sexual partners have not been interviewed on their judgement of the ring as they may greatly influence the decision of the participants to use it. This must be discussed.”

RESPONSE: The following sentence has been added to the limitations on line 319 “Sexual partners were also not interviewed and may have influenced the decision of the participants to use the ring, although participants were asked about their partner’s support.”

---

## [Editor Report · Decision Letter 1]

25 Jan 2022

Acceptability of an extended duration vaginal ring for HIV prevention and interest in a multi-purpose ring

PONE-D-21-20792R1

Dear Dr. Stoner,

We’re pleased to inform you that your manuscript has been judged scientifically suitable for publication and will be formally accepted for publication once it meets all outstanding technical requirements.

Kind regards,

R. Karl Malcolm, PhD

Guest Editor

PLOS ONE

Additional Editor Comments (optional):

Based on the authors' comprehensive responses to the referees' comments and the accompanying edits to the manuscript, I am happy to recommend acceptance of this manuscript for publication in PLOS ONE. In the interests of transparency, I participated as a reviewer for the initial evaluation of this manuscript.

---

## [Editor Report · Acceptance letter]

7 Feb 2022

PONE-D-21-20792R1 

Acceptability of an extended duration vaginal ring for HIV prevention and interest in a multi-purpose ring 

Dear Dr. Stoner:

I'm pleased to inform you that your manuscript has been deemed suitable for publication in PLOS ONE. Congratulations! Your manuscript is now with our production department. 

Kind regards, 

on behalf of

Professor R. Karl Malcolm 

Guest Editor

PLOS ONE